# The RNA Revolution in the Central Molecular Biology Dogma Evolution

**DOI:** 10.3390/ijms252312695

**Published:** 2024-11-26

**Authors:** William A. Haseltine, Roberto Patarca

**Affiliations:** 1Access Health International, 384 West Lane, Ridgefield, CT 06877, USA; roberto.patarca@accessh.org; 2Feinstein Institutes for Medical Research, 350 Community Dr, Manhasset, NY 11030, USA

**Keywords:** central molecular biology dogma, noncoding RNA, RNA modifications

## Abstract

Human genome projects in the 1990s identified about 20,000 protein-coding sequences. We are now in the RNA revolution, propelled by the realization that genes determine phenotype beyond the foundational central molecular biology dogma, stating that inherited linear pieces of DNA are transcribed to RNAs and translated into proteins. Crucially, over 95% of the genome, initially considered junk DNA between protein-coding genes, encodes essential, functionally diverse non-protein-coding RNAs, raising the gene count by at least one order of magnitude. Most inherited phenotype-determining changes in DNA are in regulatory areas that control RNA and regulatory sequences. RNAs can directly or indirectly determine phenotypes by regulating protein and RNA function, transferring information within and between organisms, and generating DNA. RNAs also exhibit high structural, functional, and biomolecular interaction plasticity and are modified via editing, methylation, glycosylation, and other mechanisms, which bestow them with diverse intra- and extracellular functions without altering the underlying DNA. RNA is, therefore, currently considered the primary determinant of cellular to populational functional diversity, disease-linked and biomolecular structural variations, and cell function regulation. As demonstrated by RNA-based coronavirus vaccines’ success, RNA technology is transforming medicine, agriculture, and industry, as did the advent of recombinant DNA technology in the 1980s.

## 1. The Classical Central Molecular Biology Dogma

In the 1860s, Gregor Mendel established the concept of particulate inheritance, in which observable traits are passed from parents to offspring through discrete hereditary information units [1]. Now known as genes, these units remain distinct and do not blend (Mendel’s law of segregation). They can be dominant or recessive with varying degrees of penetrance (now called alleles; law of dominance) and are inherited independently (law of independent assortment). In later molecular genetic studies on bacteriophages, double-stranded DNA molecules contained genes with the information to produce proteins (reviewed in [2]).

In the classical central molecular biology dogma describing the fundamental colinear and irreversible flow of genetic information within biological systems published by Francis Crick [3,4,5], information encoded in double-stranded DNA, transcribed into RNA, and translated into protein [6] determines phenotype, i.e., structures, functions, and signals, at cellular and organismal levels (Figure 1).

In the central molecular biology dogma, DNA, via its replication, is the sole universal vehicle for vertical transmission of genetic information. The DNA molecule encodes information for cellular processes and functions, with genes serving as information units. RNA is a primarily linear messenger intermediary between the information transcribed from DNA and, after ribosome-mediated translation, that in proteins. The differential timing of gene expression determines cell lineage [6].

The central molecular biology dogma is universal. All cell- and capsid-based organisms on Earth have the same building blocks for nucleic acids, proteins, lipids, carbohydrates, and metabolites. Similar to building with LEGO blocks, the construction of a multitude of structures requires a relatively small number of building blocks for each biomolecule. Growing structural complexity requires more complex instructions rather than more building block options. In some cases, structures require new building block configurations obtained via modification of existing ones.

## 2. The Flow of Genetic Information Is Not Solely Colinear and Irreversible from DNA-to-RNA-to-Protein-to-Phenotype

Although foundational and still valid, the classical central molecular biology dogma has gradually expanded thanks to discoveries over more than six decades. The dogma initially described a unidirectional, irreversible, colinear flow of genetic information from DNA to RNA to protein. In the expanded dogma, all biomolecules intricately intertwine in a multidirectional flow of genetic information to determine phenotype, with RNA predominantly underlying biological diversity.

### 2.1. RNA Can Store Vertically Transmitted Genetic Information and Serve as a Template to Generate DNA

According to the classical central molecular biology dogma, genetic information is encoded in DNA and transmitted through DNA-templated DNA replication. However, protein- and non-protein-catalyzed RNA-templated RNA replication allows the transfer of RNA as hereditary material [7].

Influenza, coronaviruses, and dengue viruses, among others, store their genetic information in RNA instead of DNA and replicate using specialized RNA-dependent RNA polymerases [8], which probably originated from ancestral transfer RNA genes [9].

Retroviruses, such as the pandemic human immunodeficiency virus (HIV)-1 causing AIDS, reverse the typical flow of information from DNA to RNA by converting their RNA genomes into DNA using an RNA-dependent DNA polymerase or reverse transcriptase [10,11] and integrating it into the host cellular genome for replication. Eukaryotic genomes via retrotransposons, referred to as the “dark genome” or the “retrobiome”, and prokaryotes use reverse transcriptase to protect against viral infection [12]. Nearly half of human DNA comprises these ancient virus-like retrotransposons that reproduce rapidly and influence aging and cancer [13,14].

Viroids are infectious single-stranded circular RNAs in plants that self-replicate without needing proteins or DNA [15]. Virusoids are satellite, non-self-replicating, single-stranded RNAs that require a helper virus to establish an infection. In plants, virusoid genomes do not code for proteins and replicate the virusoid [16], while some animal virusoids, such as viral hepatitis D, encode proteins [17].

### 2.2. Most RNAs Do Not Encode Proteins

After the formulation of the classical central molecular biology dogma, transfer RNAs, ribosomal RNAs, ribozymes, riboswitches, and the spliceosome, among others that form secondary structures of hairpin loops and bulges, were characterized and ascribed non-protein-coding functions in the formation and function of ribosomes and RNA catalytic and gene expression functions. For instance, transfer and ribosomal RNAs do not encode proteins but play a crucial role in their synthesis by helping translate messenger RNA codons into amino acids and forming the ribosome core structure, respectively [18].

Noncoding RNAs interact with the cellular proteins, metabolites, and other nucleic acids to regulate processes, such as gene expression, chromatin remodeling, epigenetic modifications, as well as cellular growth, proliferation, and survival. Most disease risk variants do not lie in coding DNA but in the less well-conserved, noncoding genomic regions, where the same DNA sequence can function differently depending on the cell type and external stimuli. Moreover, noncoding genes may lie far from the disease-causing genes they regulate, rendering it challenging to identify and mechanistically resolve disease risk loci [18]. To this end, long-read sequencing of DNA revealed even greater complexity in variants than traditional sequencing of shorter segments had documented [19].

Messenger and noncoding RNAs display an array of functionally relevant secondary and tertiary structural features across cell- and capsid-based organisms, with each primary sequence able to attain many secondary and tertiary structures [20,21,22,23,24,25,26,27,28]. Over time, it also became apparent that the vast majority of RNA made in cells, over ninety-five percent of the transcriptional output of eukaryotic genomes, is non-protein-coding RNA [29], vastly exceeding that of protein-coding RNAs. Including inherited changes in noncoding RNAs increases the number of inherited genes by an order of magnitude or more.

A vast layer of regulatory noncoding RNAs constitutes most of the genomic programming of multicellular organisms controlling epigenetic trajectories [30]. Various small and large noncoding RNAs regulate almost every gene expression level, activating and repressing homeotic genes, targeting complexes that remodel chromatin and underlying differentiation and developmental processes in simple and complex eukaryotes [31]. As the complexity of multicellular organisms increases, so does the prevalence of non-protein-coding genes, unlike protein-coding genes, which remain relatively static.

Figure 2 summarizes the various types of noncoding RNAs. Occasionally, coding RNA genes can also encode noncoding RNAs, and vice versa, noncoding RNA genes can encode proteins.

### 2.3. Biological Diversity Primarily Results from RNA Generation, Processing, and Regulation Complexities

RNA diversity underlies most intra- and interspecies diversity, far exceeding diversity associated with DNA structural and functional complexities. RNA diversity is generated via the combinatorial effects of various mechanisms, including noncoding gene expression, alternative transcription start sites, alternative splicing, splicing regulation, and alternative polyadenylation. The latter features vary among cell types, influencing protein architecture and disease-linked variation [32,33].

#### 2.3.1. Alternative Splicing

Alternative splicing of exons contributes to tissue-specific gene regulation in over 90% of multiexon genes [34]. In the complex human brain with extensive co- and post-transcriptional gene regulation [35,36,37], alternative splicing increases transcript diversity, influencing differential brain region and cell type development, function, plasticity, and neuropsychiatric pathogenesis [32,38].

Introns can also play a role in pathogenesis. For instance, persistent expression of HIV-1 intron-containing RNA in macrophages contributes to chronic immune activation, aberrant inflammation, and T-cell dysfunction even with effective combined antiretroviral therapy [39,40].

#### 2.3.2. Alternative Polyadenylation

Alternative polyadenylation introduces distinct termination sites and regulates 3′-untranslated regions, allowing various means of gene expression regulation, including microRNA-mediated translation control, RNA localization, and turnover [41].

#### 2.3.3. Regulatory RNA-Binding Proteins

Regulatory RNA-binding proteins, numbering between two and three thousand in humans [42,43], differentially influence RNA processing based on their affinities for different RNA sequences and the local abundances of RNAs and proteins [44]. In brain cells, for instance, driven by RNA-binding proteins, individual transcripts can adopt various structures, thereby defining cellular identities [45]. RNA binding proteins also contribute to the dynamic regulation of many RNAs involved in spermatogenesis [46].

RNA-binding proteins assemble coding and noncoding (including long, ribosomal, and transfer) RNAs and other proteins into ribonucleoprotein complexes [47,48,49]. These eukaryotic gene expression post-transcriptional and translational regulator complexes participate in diverse RNA activities, such as export, splicing, stability, translation, and degradation [50,51,52].

Some proteins bind to both DNA and RNA, as is the case of the topoisomerase I enzyme (TOP1). TOP1 prevents genomic instability by alleviating DNA torsional strain. TOP1 introduces transient single-strand breaks that prevent supercoiling and torsional stress accumulation, which could otherwise lead to damage and instability of DNA and cell death [53]. TOP1 is also an RNA-binding protein. Interactions between RNA and TOP1 regulate DNA during transcription by modulating TOP-1-mediated relaxation. In cancer cells, for instance, DNA transcription is often elevated, necessitating increased levels of TOP1 activity to relax the DNA and maintain proper gene expression. RNA opposes TOP1 activity. Inhibiting RNA binding of TOP1 may work similarly to TOP1 inhibitors like camptothecin by increasing TOP1 catalytic complexes on DNA [53].

#### 2.3.4. Formation of Ribonucleoprotein Complexes

Ribonucleoprotein complexes dynamically remodel to adapt to cellular needs and environmental conditions [54]. Elucidating their composition and function warrants further study at the cell, tissue, and organismal levels [50]. For instance, the microglial-secreted complement protein C1q is internalized by neurons in an age-dependent manner, undergoes RNA-mediated interactions with neuronal ribonucleoprotein complexes, and alters neuronal protein translation and homeostasis in the adult and the aging brain, highlighting how temporally regulated neuroimmune interactions impact critical intraneuronal functions [55].

#### 2.3.5. Formation of Biomolecular Condensates Through Liquid–Liquid Phase Separation

Biological macromolecules can phase-separate in the cell to form assemblies in which proteins and nucleic acids accumulate in high concentrations in biomolecular condensates [56]. Condensates modulate interactions and chemical reactions at the molecular scale, organize biochemical processes at the mesoscale, and compartmentalize cells, playing a pivotal role in ribosome assembly, RNA splicing, stress response, mitosis and chromatin organization, and other cellular processes and diseases. Such interactions are particularly prevalent for intrinsically disordered proteins, which either lack a well-defined three-dimensional structure or contain large, disordered regions that can mediate interactions with several binding partners [56].

Kdm1a is a histone demethylase that remains highly expressed in the adult brain and is linked to intellectual disability with essential roles during gastrulation and the terminal differentiation of specialized cell types, including neurons [57]. Kdm1a’s amino terminus contains an intrinsically disordered region essential to segregate Kdm1a-repressed genes from the neighboring active chromatin environment. Segregation of Kdm1a-target genes is weakened in neurons during natural aging, underscoring the role of Kdm1a in safeguarding neuronal genome organization and gene silencing throughout life [57].

Liquid–liquid phase separation and related phase transitions have emerged as a biophysical mechanism for nuclear compartmentalization assembly in distinct liquid-like structures that participate in transcriptional control and other chromatin-related functions [58]. Albeit sharing the same genetic material, each cell type in a multicellular organism expresses a different set of genes and accomplishes distinct functions. This is achieved through epigenetic mechanisms that regulate the accessibility and packaging of the genetic material inside the cell nucleus [57]. Chromatin has been classically classified into euchromatin and heterochromatin, respectively linked to active transcription and gene silencing [59]. On a smaller scale, the genome is organized in topologically associating domains [60,61], where extensive chromatin interactions are observed. The basic units of chromatin interactions are chromatin loops involving two looping anchors [62,63] and are mediated by specific protein factors [64]. Many proteins, such as the CCCTC-binding protein [63,65], help establish and maintain the overall chromatin folding architectures that provide the genome-wide gene expression framework underlying cell identity.

Whether ubiquitous, cell type-specific, or stress-induced, RNA granules are intracellular RNA-protein assemblies or condensates not enclosed by membranes, ranging from ∼100 nm to several micrometers in diameter and housing RNA-focused activities [66,67]. There are over 20 RNA granule types, each with a unique composition, sometimes comprising dozens of proteins and thousands of RNAs [66,67,68,69,70,71]. RNA granules assemble by phase separation of subsoluble ribonucleoprotein complexes that partially phase-separate from the cytoplasm or nucleoplasm [67].

RNA granule proteins function in RNA metabolism, from nuclear transcription and processing to cytoplasmic translation and RNA turnover. For example, as the primary cellular compartment for ribosome biogenesis, nucleoli assemble around nascent ribosomal RNAs and concentrate ribosomal proteins and assembly factors [72]. Many RNA granules have been assigned putative functions based on composition, including P-bodies as mRNA storage or decay sites and nuclear speckles as mRNA splicing sites [73,74,75]. mRNAs’ localization in stress granules may prevent their translation during stress [76].

In neurons, RNA granules in the soma and cellular processes contain transcripts encoding signaling, synaptic, and RNA-binding proteins linked to neurological disorders as pathological inclusions [77]. RNA granules coupled to lysosome-related vesicles in the axon carry both microRNAs [78,79] and messenger RNAs [80] for regulated local translation at sites distal from the soma, which is critical for maintaining axonal homeostasis and avoiding axonal degeneration [81]. Many components are shared with HIV-1 ribonucleoprotein trafficking granules [82]. Moreover, axonal late endosomes serve as platforms for the local translation of nuclear-encoded mitochondrial messenger RNAs by associated ribosomes [83].

#### 2.3.6. RNA Modifications

Many RNA modifications have been identified, including N^6^-methyladenosine (m^6^A), N^1^-methyladenosine (m^1^A), 5-methylcytosine (m^5^C), N^7^-methylguanosine (m^7^G), N^6^, 2′-O-dimethyladenosine (m^6^A_m_), and N^4^-acetylcytidine (ac^4^C), among others [84,85]. Methylation of N^6^-methyladenosine (m^6^A) is the most prevalent internal and reversible modification of RNAs [86] and an important regulator of RNA biology [87]. RNA methylation directly impacts protein production to achieve quick modulation of dynamic biological processes [87]. RNA’s m^6^A modification regulates gene expression by affecting chromatin packaging, thereby directing which stretches of DNA are expressed into proteins. m^6^A modifications lead to different outcomes, such as decay, stabilization, or transport of RNAs. Cell types differentially display their unique m^6^A profiles determined by m^6^A writers and erasers. m^6^A readers and their interacting proteins interpret the m^6^A-encoded epigenetic information. m^6^A readers participate in various biological processes, including regulating fate transitions, and their defects underlie diverse diseases [86].

Messenger RNA length correlates with its enrichment in stress granules, with long mRNAs often containing one or more long exons, which are preferential sites of m^6^A formation [76]. m^6^A RNA methylation also regulates mitochondrial function by promoting polysome-associated nuclear-encoded mitochondrial complex subunit RNA translation, and its dysfunction underlies neurodegenerative disorders [87].

#### 2.3.7. Regulatory RNAs or Riboregulators

Regulatory RNA molecules, or riboregulators, including long noncoding RNAs and microRNAs, regulate gene activity across almost all levels of biological systems, including wound healing, chromatin arrangement, transcription, suborganelle stabilization, and posttranscriptional modifications [88,89]. In bacteria, regulatory RNAs respond to changes within their microenvironments and several-control virulence [90].

Regulatory RNAs of different sizes, shapes, and functions utilize several export pathways from the nucleus to the cytoplasm through nuclear pore complexes via export receptors and adaptor proteins [91]. Mex67-Mtr2/NXF1-NXT1 is the principal export receptor for bulk mRNA export [92,93]. The Crm1/Xpol karyopherin supports export through interaction with the cap-binding complex attached to the 5′ end of transcripts [94]. Both pathways export some noncoding RNAs, like long noncoding RNAs and small nuclear RNAs, respectively [94]. Some regulatory RNAs, like certain microRNAs, may use specialized export pathways like the RNA-induced silencing complex [89].

Several factors can promote the export of regulatory RNAs, such as being double- vs. single-stranded [94], proper RNA processing, including splicing and 5′ capping [92], and certain sequence features, such as GC-rich regions, by recruiting specific export factors [92].

Several factors can limit or regulate the nuclear export of regulatory RNAs. Understanding this has important implications for gene regulation [89], pathogenesis [91], and therapeutics.

The cell nucleus actively retains some regulatory RNAs. For example, long noncoding RNAs like Xist are primarily nuclear and function within the nucleus [89]. The cell employs various quality control mechanisms to prevent the export of improperly processed RNAs. For instance, intact 5′ splice site motifs can promote nuclear retention of misprocessed messenger RNAs and some long noncoding RNAs [92]. The export machinery can become saturated, potentially limiting the export rate of specific RNAs, especially under stress conditions. Some regulatory RNAs can modulate the export of others. For example, specific antisense RNAs can influence the export and expression of their sense counterparts [94]. The efficiency of regulatory RNA export can vary depending on cell type and physiological conditions. For instance, stress can alter export dynamics for specific transcripts [91].

#### 2.3.8. RNA Stability

Factors influencing proteins’ half-lives and context-dependent variations in proteins’ repertoires affect RNA’s stability. For instance, mRNA’s stability is influenced by polyadenylation, which introduces a 3′-poly-A tail; capping, which adds a methylated guanosine residue to the 5′ end; splicing; and nuclear exit of messenger RNAs, among others warranting further study.

Messenger RNA poly(A) tail length changes strongly influence precise temporal regulation of translation after transcription ceases during oocyte maturation and early embryogenesis in vertebrate animals, with longer-tailed messenger RNAs translated more efficiently [95].

In muscle cell formation, the RNA-binding proteins YB1 and HuR form a heterodimer that associates with a U-rich consensus motif to stabilize key promyogenic mRNAs [96,97].

Cys2His2 zinc finger proteins, the largest group of over 700 DNA-binding factors for transcription via promoter regions, also regulate post-transcriptional processes that modify messenger RNAs by binding to their 3′-untranslated regions. Cys2His2 zinc finger proteins regulate precursor messenger RNA splicing, length through cleavage and polyadenylation, and function via m^6^A modification [98,99]. Transcripts with long 3′ untranslated regions are often less stable and degraded more rapidly by cellular enzymes, while short-tailed ones are more protected from degradation. A Cys2His2 zinc finger protein, SP1, acts to yield shorter 3′ untranslated region tails. Untranslated region trimming is accomplished by the RNA cleavage machinery, which snips the RNA near where Sp1 is bound [100].

Messenger RNA is degraded when no longer needed. In eukaryotic cells, messenger RNA decay is catalyzed by two pathways initiated by deadenylation of the polyadenylated (poly-A) tail. After decapping, 5′ to 3′ RNA degradation is accomplished by Xrn1, a 5′ to 3′ exoribonuclease. In the 3′ to 5′ pathway, RNA degradation is catalyzed by a multi-subunit 3′ to 5′ exoribonuclease complex, the RNA exosome. The exosome also processes small nucleolar, small nuclear, and ribosomal RNAs. It degrades many noncoding RNAs, including those arising from bidirectional transcription or enhancers to maintain RNA homeostasis in eukaryotic cells.

The various molecular machines that translate and degrade messenger RNA are physically linked to each other, forming a super complex encompassing the ribosome, the SKI complex, and the exosome. The RNA exosome is a versatile RNA-degradation machine involved in RNA maturation, RNA surveillance, regulation of messenger RNA levels, and processing of various stable RNA species [101]. As a primary mode, the exosome participates in RNA degradation via its 3′ to 5′ exoribonuclease activity. RNA is degraded from the 3′ end, and in eukaryotes, endoribonuclease activity cleaves RNA internally. The exosome recognizes specific RNA features via specificity factors and recruits activating complexes, including helicases that unwind RNA to facilitate degradation.

The exosome complex operates in messenger RNA turnover and degradation of aberrant transcripts in the cytoplasm, the processing of ribosomal, small nucleolar, and small nuclear RNAs in the nucleus, and ribosomal RNA maturation in the nucleolus [102]. The exosome is crucial for RNA quality control, degrading improperly processed messenger RNAs, eliminating cryptic unstable transcripts, and removing aberrant RNA molecules that fail to mature correctly [103].

### 2.4. RNA Can Guide Other Molecules That Modify DNA, Regulating Gene Expression

While RNA does not directly bind to DNA to regulate it in the same way proteins like transcription factors do, in RNA-directed DNA methylation in plants, for instance, noncoding RNAs direct the addition of methyl groups to specific DNA sequences, leading to the silencing of genes involved in abiotic stress responses, development, and the suppression of transposable elements [104].

### 2.5. RNA Can Directly Affect Extracellular Biology and Pathology

#### 2.5.1. GlycoRNAs

Glycosylation not only modifies proteins and lipids. GlycoRNAs primarily present at the cell surface consist of small nuclear RNAs modified with secretory N-glycans rich in sialic acid and fucose via their attachment to the modified base 3-(3-amino-3-carboxypropyl)uridine (acp^3^U) [105,106]. These glycoRNAs in mammals and other eukaryotes interact with antibodies and cellular receptors, influencing neutrophil recruitment, immunity, and pathogenesis [105,106,107,108,109,110]. GlycoRNAs, including glycosylated transfer, ribosomal, small nuclear, small nucleolar, and Y noncoding RNAs, could explain why RNAs, traditionally considered intracellular molecules, act as autoantigens [107]. Moreover, the queuosine modification of transfer RNAs affects translation elongation rates, codon recognition, and biofilm formation and virulence in bacteria.

#### 2.5.2. RNAs Can Be Transferred Intra- and Inter-Species in Extracellular Vesicles

RNA is constantly responding to dynamic conditions inside and outside the cell. RNAs can be transferred via extracellular vesicles to warn other cells about pathogens or affect their metabolic status and genetic expression profile [111]. RNA’s flexible backbone allows it to fold into several shapes that can impact cell biology. RNA can act as an enzyme to accelerate cellular chemical reactions. It can bind to DNA to activate or silence gene expression. Competing strands of RNA can tangle up messenger RNA instructions during RNA interference, preventing the production of new proteins.

Transcending taxonomic barriers, organisms from different kingdoms exchange RNA. For instance, the *Botrytis cinerea* fungus, a fuzzy gray mold that affects strawberry and tomato crops, delivers RNAs in extracellular vesicles that interfere with the *Arabidopsis* (thale-cress) plant’s ability to fight the infection [112]. Plant cells respond with RNAs in extracellular vesicles that damage the fungus [113,114]. Parasitic worms in mouse intestines release RNA in extracellular vesicles that quell the host’s defensive immune proteins [115]. Bacteria can do likewise in human cells [116]. The fungus *Candida albicans* uses human RNA in extracellular vesicles to promote its growth [117]. In a symbiotic relationship, bacteria living in the roots of legumes export RNA to promote nodulation, fixing nitrogen for the plant [118].

### 2.6. Prions Are Infectious Proteins That Transmit Genetic Information Without DNA Mediation

Prions add another layer of complexity beyond the classical central dogma of genetic information flow by converting normal proteins into the infectious prion form. A prion, a misfolded rogue form (PrP^SC^) of a normal cellular protein (PrP^C^), may result from a genetic mutation, occur spontaneously, or be infectious. PrP^SC^ stimulates other endogenous normal proteins to become misfolded via a crystallization-like transformation, generating fibrils that cluster into plaques. These plaques underly various forms of transmissible spongiform encephalopathy that in humans include Creutzfeld–Jacob disease, kuru, Gerstmann–Sträussler–Scheinker syndrome, and fatal insomnias [119].

Prions vary in size and shape, with 3-dimensional configurations influencing pathogenicity and conferring strain-like properties [119]. More recently, electron cryo-microscopy mapping of fibrils from the brains of mice and hamsters revealed extra densities on fibrils with a Y-shaped polymer consistent with RNA, possibly entailing a short tandem repeat [120]. This may also apply to fibrils in Alzheimer’s disease, amyotrophic lateral sclerosis, and other neurodegenerative disorders. RNAs with differing base sequences could exert a specific tethering effect on the particular protein shape found in each disease [121] or exert direct toxicity as implicated in some neurodegenerations [122]. Therefore, prions may not only consist of protein but constitute a new type of infectious agent in which RNA might be able to regulate or replicate [120].

### 2.7. Proteins Can Regulate DNA Gene Expression via Transcription Factors and Histone Modification, Underlying Epigenetic Inheritance Beyond Genomic DNA Inheritance

The human genome consists of six and a half feet (two meters) of DNA packed inside the cell in a nucleus barely 10 μm in diameter, or 100,000 times smaller than the length of the genome’s DNA. Most of the genetic material of eukaryotic cells is stored in the nucleus as chromatin, a nucleoprotein complex comprising DNA, histones, and other structural and regulatory factors. For instance, the protein complex, including the Mre11 protein, plays a pivotal role in maintaining the stability of the genome by sensing and repairing double-strand breaks in DNA. Alterations to Mre11 via mutations inherited or developed during life and its methylation, SUMOylation, and phosphorylation can lead to oncogene activation and cancer by chronic innate immune system stimulation [123].

Mitochondria contain genetic information, and a dedicated translation system is used to express it [124]. However, higher-order organisms have progressively smaller mitochondrial genomes, reflecting the translocation of mitochondrial genes into the nuclear genome over evolutionary time, facilitating the coordinated synthesis of organellar proteins by the cytosolic translational machinery [125]. Likewise, mitochondrial DNA genetic variations in adult blood can decline with age, even becoming undetectable consonant with the higher prevalence of mitochondrial disease in children via inheritance [126].

Throughout their lifespan, cells must continuously activate and deactivate genes. This process is regulated by transcription factors, a class of proteins that control gene expression by binding to DNA sequence motifs in a chromatinized genome, where nucleosomes can restrict DNA access to DNA. Some transcription factors can displace nucleosomes when binding at specific regions [127]. Transcription factors, differentially expressed among cell types, bind to and modulate genomic functional areas. As DNA sequences that drive cell-type-specific gene expression, developmental transitions, and cellular responses to external stimuli [128,129,130,131], enhancers are bound by transcription factors that signal to other factors about whether to express a target gene, with several properties, including transcription factors binding affinity, cooperative binding and clustering, specifying enhancer function [132,133,134,135].

A transcription factor subset, the pioneer factors, synergize with other transcription factors to target nucleosomal sites in closed chromatin by binding specific sequences and interacting with histones. This differential transcription factor synergy contributes to differential gene expression among cell types, and sequence variation in enhancers contributes significantly to phenotypic variation within populations [136].

Beyond transcription factors’ binding, the tightness with which a segment of DNA winds around histones affects the expression of the genes it contains; tightly wound DNA tends to restrict gene activity, whereas loosely wound DNA allows genes to be more active. This also affects processes such as memory formation and retention. In a study on mice, neurons with less-compressed DNA were more likely to be recruited into the set of cells, known as an engram, that form and retain new memory.

Chromatin-associated retrotransposon RNA 5-methylcytosine (m^5^C) can be recognized by the methyl-CpG-binding-domain protein MBD6, which guides deubiquitination of nearby monoubiquitinated Lys119 of histone H2A (H2AK119ub) to promote an open chromatin state [137]. TET2 oxidizes m^5^C and antagonizes this MBD6-dependent H2AK119ub deubiquitination. Tet2 is more active in stem cells during their differentiation, and cancers and other pathologies involve TET2 overproduction-related mutations [137].

Beyond the modifications mentioned, histones can undergo lactylation, citrullination, crotonylation, succinylation, SUMOylation, propionylation, butyrylation, 2-hydroxyisobutyrylation, and 2-hydroxybutyrylation, involved in transcription, replication, DNA repair and recombination, metabolism, chromatin structure, and promoting the occurrence and development of various diseases, with clinical applications as therapeutic targets and potential biomarkers [138].

Epigenetic modifications, including DNA and histone modifications, combine into characteristic patterns that demarcate functional regions of the genome, such as enhancers, promoters, gene bodies, and heterochromatin, and control gene expression [139,140,141].

It is broadly accepted that the inheritance of genomic DNA mainly drives biological inheritance. However, trans- and intergenerational epigenetic inheritance is present in bacteria, protists, fungi, plants, and invertebrate and vertebrate animals [80,142,143,144,145,146,147,148,149,150,151]. For instance, the human genome can be methylated in about 30 million locations, known as cytosine-phosphate-guanine (CpG) sites. The modifications often turn genes on or off, and their distribution across the genome changes as people age. DNA methylation of promoter-associated CpG islands can be transmitted from parents to their offspring in mice [152]. Animals transmit DNA and other molecules, such as RNA, proteins, and metabolites, to their progeny via gametes [151]. Moreover, in *Caenorhabditis elegans* nematodes, information on environmental responses is transmitted across generations through RNA-dependent RNA polymerase-amplified small RNAs [150]. Regulatory small RNAs enable sequence-specific gene regulation and, unlike chromatin modifications, can move between tissues and escape from immediate germline reprogramming. Inheritance of small RNAs could spread adaptive ancestral responses [150]. Therefore, there is more to heredity across evolution than simply genomic DNA inheritance.

### 2.8. Genetic Mosaicism

Initially, it was believed that DNA in all cells of an organism is identical. However, mutations arising during mammalian cell division become fixed in a daughter cell and are passed on to subsequent cell progeny, potentially giving rise to variant cell clones. Multiple cell types and tissues can have the same variants if acquired during early embryogenesis. Consequently, mammals are complex mosaics of genetically distinct clones. This so-called mosaicism is more commonplace with aging, possibly affecting tissue function and disease development [153]. Insertions of retrotransposons in different genomic regions are associated with mosaicism [154].

## 3. One Gene Can Encode Multiple Proteins and Noncoding RNAs

A gene is currently defined as one or more segments of DNA (or RNA in some viruses or virusoids) that are not necessarily physically contiguous, code for one or more functional proteins and noncoding RNAs, and are evoked by and participate in gene regulatory networks via one or more promoters and distal regulatory elements. Protein-coding genes tend to be more conserved across species, while diverse and species-specific noncoding genes contribute significantly to biological diversity.

Based on the principles of Mendelian inheritance and studies on bacteriophages, the classical central dogma of molecular biology was formulated on the premise that each gene encodes only one protein. However, it was later realized that one gene can encode multiple proteins, noncoding RNAs, and occasionally both.

### 3.1. Splicing

A gene can be split into exons, i.e., protein-coding non-immediately contiguous segments separated by introns, and assembled via splicing with differential exon representation, i.e., alternative splicing [155]. Therefore, with only about 20,000 protein-coding genes in the human genome [156], there are over 100,000 proteins. Over 90% of multiexon human genes undergo alternative splicing [157], resulting in hundreds of thousands of protein isoforms (variants) [158,159,160]. Underlying diversity in protein open reading frame and function, isoforms differ in coding sequence via exon skipping, a choice between mutually exclusive exons, alternative splice site use, intron retention, and introduction of ‘poison exons’ affecting functional full-length protein levels [161,162]. As an additional diversity source, different 5′- and 3′-untranslated regions quantitatively affect cellular functions, including triggering human diseases [157,163,164].

### 3.2. Translation from Noncanonical Open Reading Frames or Generation from Noncoding RNAs

Peptides can be translated from noncanonical open reading frames, blurring the distinction between coding and noncoding genes [165,166]. For instance, in dual RNA, telomerase RNA, which is a long noncoding RNA essential for maintaining chromosome stability and cellular immortality in eukaryotes, is processed from a messenger RNA transcript that encodes a conserved protein [167]. Moreover, the long noncoding mammalian telomeric RNA (TERRA) can generate two dipeptide repeat proteins, repeating valine-arginine or glycine-leucine. Their abundance could alter nucleic acid metabolism and general protein synthesis and trigger cellular inflammation responses [168].

Genes free-floating outside of the traditional organismal genome and encoding proteins can be synthesized from noncoding RNAs [169]. As part of their defenses against bacteriophages [170,171,172], bacteria use retrons, composed of reverse transcriptase and a noncoding RNA, to generate fully functional, free-floating, transient genes essential for survival but not preserved in the genome [29]. The noncoding RNA and reverse transcriptase enzyme are constitutively expressed in uninfected cells from a single promoter. This leads to synthesizing a repetitive single-stranded cDNA via precise, programmed template jumping that mediates rolling circle reverse transcription. Phage infection triggers second-strand synthesis, leading to the accumulation of double-stranded, concatemeric cDNA molecules. A promoter created across the junction between adjacent cDNA repeats then leads to the expression of abundant, heterogeneously sized mRNAs encoding a stop codon-less, *n*early *e*ndless *o*pen reading frame (*neo*) gene. The *ne*o protein restricts viral infection via potent cell growth arrest activity [29].

The *neo* protein results from complex and repeated transitions back and forth between DNA- and RNA-based carriers of genetic information before translation finally yields a protein product. A bacterial version of reverse transcriptase reads RNA as a template to make entirely new genes written in DNA. These genes are then transcribed back into RNA, which is translated into protective proteins when a virus infects a bacterium. By contrast, viral reverse transcriptases do not generate new genes; they merely transfer information from RNA to DNA. This finding further challenges the universal paradigm that genes are encoded linearly along a chromosome. Genes in living organisms are arranged from head to tail, even with splicing in eukaryotes. Along with strategies to compactly encode genetic information, including ribosomal frameshifting, overlapping open reading frames, and nested genes, these findings add another layer of complexity to how protein-coding sequences can be stored in a genome [29].

*Neo* genes are hidden in regions of genomes previously thought to be exclusively non-coding. Alternative approaches are needed to discover genes like *neo* in eukaryotes and other bacterial reverse transcriptases that could eventually constitute part of therapeutic gene editing systems [29]. To this end, prokaryotic reverse transcriptases, which together with a noncoding RNA result in the production of a toxic protein [12], appear to have a common retroelement ancestor, the Group II intron, which catalyzes self-splicing and site-specific DNA insertion and is also thought to be the precursor to the eukaryotic spliceosome [173].

### 3.3. Stop Codon Readthrough

Stop codon readthrough during translation endows a protein with extended carboxyl terminus regions with new functions. For instance, ~30% of the transcripts of the human water channel *Aquaporin-4* gene have an extra carboxyl-terminus piece that allows the extended Aquaporin-4 protein to localize to the perivascular or Virchow–Robin space surrounding small blood vessels in the brain. This property maintains the appropriate fluid volume in said space for normal cognitive function and facilitates amyloid clearance [174]. A similar stop codon readthrough mechanism yields variants of opioid receptors [175].

### 3.4. Generation of Microproteins or Short Bioactive Peptides

Microproteins are encoded from short open reading frames three to fifteen amino acids long [176]. This new proteome component affects translation, endocytosis, cell survival, brain development, morphogenetic development, and cancer. Moreover, short bioactive peptides cleaved from bigger precursor proteins can function as peptide hormones [176].

### 3.5. Overlapping Reading Frames or Genes

The three reading frames in a gene could generate different proteins [177]. For instance, the capsid (C) protein of henipaviruses is produced by leaky scanning of ribosomes during translation. The scanning ribosome bypasses the initial AUG start codon and begins translation further downstream, resulting in the use of an alternate open reading frame within the phosphoprotein (P) gene mRNA [178].

Gene overlap (genes-within-genes) is common among viruses and bacteria, and it can express more than one protein from a single messenger RNA cistron controlled by the messenger RNA’s primary and secondary structure [179,180]. Using additional translation initiation sites within the gene is one such mechanism [181,182]. Proteins whose translation is initiated at different start sites within the same reading frame will differ in their amino termini but will have identical carboxyl-terminal segments. As a second mechanism, an alternative initiation of translation in a register different from the frame dictated by the primary start codon will yield a protein whose sequence is entirely different from the one encoded in the main frame [179].

### 3.6. RNA Processing and Modification, Including Editing

Processing and modifying the RNA encoded by one gene can yield hundreds to thousands of distinct RNAs, diversifying genetic and functional repertoire [183]. RNA editing, for instance, Adenosine-to-Inosine editing, is one such mechanism. RNA editing via the insertion or deletion of nucleotides during the transcription of an mRNA allows henipaviruses and other paramyxoviruses to produce multiple accessory proteins involved in viral pathogenicity from one gene [184]. Henipaviral V and W proteins, which antagonize the host’s interferon response, are produced via specific and conserved insertion of additional guanine residues into the phosphoprotein (P) gene mRNA during transcription [185]. This insertion causes a frameshift, resulting in the V protein sharing the amino-terminal domain with P but having a unique carboxyl-terminal domain. For the W protein, two guanosine residues are inserted at the same editing site used for the V protein. This double insertion causes a different frameshift, giving W a unique C-terminal domain compared to P and V proteins [185]. The C-terminal domains confer differential cellular localization and interferon antagonism targets. Other viruses use RNA editing, including *Filoviridae*. The Ebola and Marburg viruses produce multiple proteins from their GP gene by inserting or deleting adenosines in the GP mRNA [185,186], some *Potyviridae* (a plant virus family) [187], and other RNA viruses (influenza A, dengue, and lymphocytic choriomeningitis) [186]. The mechanism and extent of RNA editing vary among virus families. In some cases, viral RNA editing is mediated by host factors rather than viral polymerases, as in the case of some DNA viruses like Epstein–Barr virus and human cytomegalovirus [186].

As an example of mastery in RNA editing granting remarkable adaptability to environmental challenges, over 60% of nervous system-relevant RNA transcripts are edited in octopuses compared to less than 1% in humans. Octopuses have eight sensitive, semi-independent limbs, reshape their bodies by controlling internal fluids, and have their thoughts paint shapes and colors on their skin [188,189].

### 3.7. Protein Modification

Various mechanisms, including ubiquitination, glycosylation, and methylation [190], can differentially modify proteins and reprogram their function in all cells or by cell type [191].

### 3.8. Antisense Strand Transcription

The antisense (minus) strand of the gene’s DNA is also sometimes transcribed. In retroviruses and other organisms, antisense transcripts may have originated as regulatory non-protein-coding molecules that later acquired protein-coding function [192,193]. In plants, for instance, the *ENOD40* gene is a dual RNA activated during a symbiotic interaction in root nodule organogenesis. The oligopeptides encoded by *ENOD40* and the structured regions of the *ENOD40* RNA interact with different proteins in the cell to control enzymatic activities or induce the relocalization of ribonucleoproteins, respectively [194].

RNA polymerase II can bidirectionally transcribe active DNA enhancers, creating enhancer RNAs. Said transcription is divergent, i.e., it starts from nearby but not identical positions and both transcripts are rarely present in the same cell. Enhancer RNAs can be polyadenylated and have shorter half-lives than messenger RNAs and long noncoding RNAs; however, their transcription frequency approaches that of messenger RNAs [195]. Enhancer RNAs crucially regulate chromatin conformation and accessibility, histone modification, transcription activation, and gene expression via chromatin loop construction that approaches enhancers to their target gene [196,197]. Specific enhancer RNAs have been associated with carcinogenesis [198] and brain diseases [195].

## 4. The Expanded Central Molecular Biology Dogma Reflects the Interrelatedness of All Biomolecules and Phenotype

The findings summarized in this review highlight the complexity and versatility of genetic information flow, demonstrating that the central dogma, while foundational, is not absolute. Table 1 summarizes some of the prominent findings in the evolution of the central molecular biology dogma, emphasizing the significant role of RNA in genetic information flow and phenotypic diversity.

Figure 3 depicts the expanded central molecular biology dogma. RNAs contribute to phenotypes directly or by modifying themselves and other biomolecules. Likewise, proteins can function as epigenetic modifiers of nucleic acids. Accordingly, the expanded central molecular biology dogma acknowledges that different biomolecules might perform similar functions while the same biomolecule can be recruited to perform a different function. Therefore, novel phenotypes can originate either through mutations in existing genotypes or through phenotypic plasticity, i.e., the ability of one genotype to form multiple phenotypes.

Among other properties, biomolecular mimicry, moonlighting, and promiscuity reflect phenotypic redundancy and functional flexibility [199]. Genotype-to-phenotype redundancy is an intrinsic property of natural systems [200,201]. In biomolecular mimicry, the same biomolecule from different origins or molecules of dissimilar composition resembles each other’s structure. Via mimicry, pathogen-derived peptides resembling host peptides trigger autoimmunity [202]; DNA-shaped proteins from pathogens evade host defenses or regulate transcription; and various RNAs, including transfer RNA look-alike factors, contribute to multilevel gene expression regulation [203].

RNAs show extensive plasticity in secondary structure phenotypes due to their continual folding and unfolding. The same biomolecule may have multiple context-dependent functions in RNA or protein moonlighting. Differential folding or catalytic activity of an RNA could lead to distinct yet functionally interrelated phenotypes in a common biochemical pathway, as seen in ribozymes [204]. Consistently, mixtures of RNA fragments that self-assemble into self-replicating ribozymes spontaneously form cooperative catalytic cycles and networks [205]. RNA folding into structures with functions different from the most stable minimum-free-energy one also underlies RNA promiscuity of a given primary sequence [206]. Similarly, enzyme promiscuity reflects their ability to catalyze reactions other than their primary one [207].

As per the scientific paradigm of molecular crowding, biomacromolecules do not act in isolation but in crowded environments that foster their interrelatedness [208]. Therefore, the central molecular biology dogma of genetic information flow as the determinant of phenotype continues to evolve as studies on the structure and dynamics of biomolecules in cell-mimicking environments and other systems garner further knowledge. Moreover, combinatorial explorations of currently available molecular variants could yield novel phenotypes and further molecular complexity subject to natural selection [199].

In conclusion, the central molecular biology dogma evolved from a colinear one in which every cell in our body contains the same genetic information encoded in DNA with differential protein expression, encoded in messenger RNA transcripts, giving cells their distinct functions. In the expanded dogma, genetic information flow is multidirectional among biomolecules, determining phenotypes at the cellular, organismal, and population levels, and organisms show mosaicism in genetic information. RNA is the predominant phenotypic diversity and regulation determinant. This allows one phenotype to derive from many genotypes and one genotype to express a variety of phenotypes, thereby preserving essential functions.

Figure 4 summarizes the contributions of RNA to genetic information flow and biological diversity.

The RNA revolution is increasing our understanding of cellular function diversity. For instance, deep RNA sequencing of individual nerve cells has distinguished 16 neuron types involved in the human sense of touch, similar among mice, macaques, and humans [209]. The interdependence among all biomolecules, including nucleic acids, proteins, polysaccharides, lipids, and metabolites, warrants further study to better design RNA-based technology for medicine, agriculture, and industry. Improvements in RNA design and delivery technologies for achieving cell type and organ specificity and minimizing off-target effects will increase the effectiveness of RNA-based approaches. The same is true for improving stability, translation, delivery, and immunogenicity avoidance of RNA-based interventions.

Emerging RNA-based approaches are or have the potential to transform human and veterinary medicine, agriculture, and industry, as did the advent of recombinant DNA in the 1980s, creating and disrupting novel diagnostics, preventive approaches, and therapeutics, as evinced by the success of the RNA-based coronavirus vaccines in the recent pandemic.

Applications in the human and veterinary medicine front include RNA vaccines, including RNA-based domestic animal vaccines for microbial and pest resistance; silencing RNAs; RNAs for gene therapy, including gene knockout and knockdown; RNA as CRISPR-Cas guides and regulators of gene expression; RNAs as antibody and protein therapeutic delivery vectors; RNA aptamers; RNA-based screening, surveillance, diagnostics, prognostication, and personalized medicine; and RNAs for faster growth of food animals, improved animal breeds, and aquaculture.

Agricultural applications include silencing RNAs and messenger RNA modifications to enhance crop yields; RNA interference to introduce favorable traits, such as non-allergic nuts and peanuts, decaffeinated coffee, nutrient-fortified grains, and longer vegetable and fruit shelf life and decreased oxidation when cut; and RNA technologies to increase resistance to pests, drought, salinity, and temperature.

Industrial uses of RNA include RNA-based synthetic biology for the production of biofuels by microorganisms, including RNA interference to favor biofuel production pathways in microbes and RNA devices to sense biofuel precursors and trigger biofuel production pathways; manufacturing of biodegradable plastics to substitute thermoplastics with environmental benefits, including use of bio-based monomers to produce bioplastics with fermentation technology; manufacturing of biodegradable polymers for drug delivery and tissue engineering, and bioscaffolds for in vitro cell culture, organoids, and in vivo implants; and RNA technology for bioremediation of the environment. However, accessing RNA technology in low- and middle-income countries remains challenging.

## Figures and Tables

**Figure 1 ijms-25-12695-f001:**
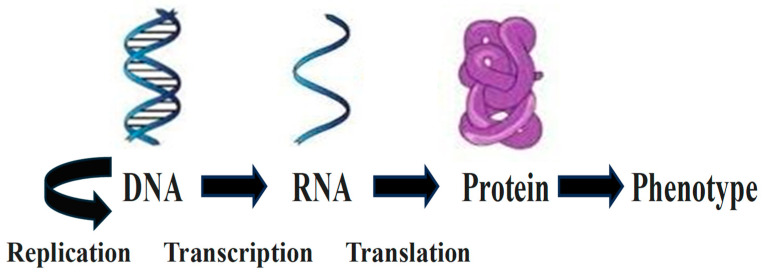
Classical central molecular biology dogma.

**Figure 2 ijms-25-12695-f002:**
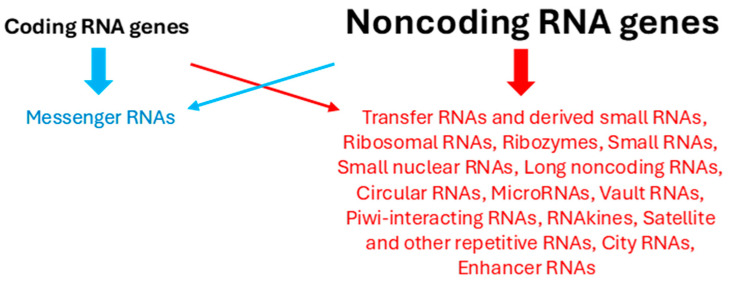
Coding (in blue) and noncoding (in red) RNAs. Occasionally, their genes can also encode the other RNA type (crossed arrows). Noncoding RNA genes vastly outnumber coding genes.

**Figure 3 ijms-25-12695-f003:**
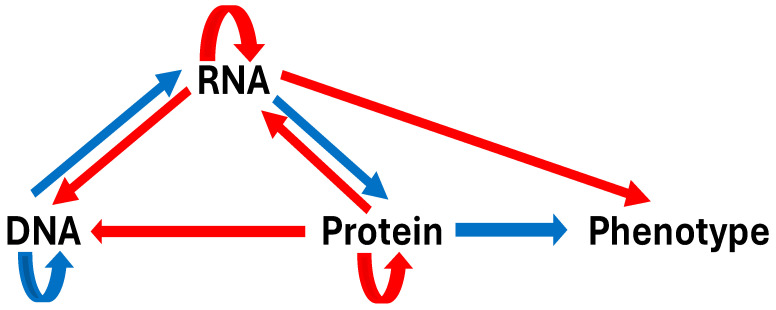
Expanded central molecular biology dogma. The initial dogma is shown using blue arrows. Subsequent findings underlying the red arrows are summarized in Table 1.

**Figure 4 ijms-25-12695-f004:**
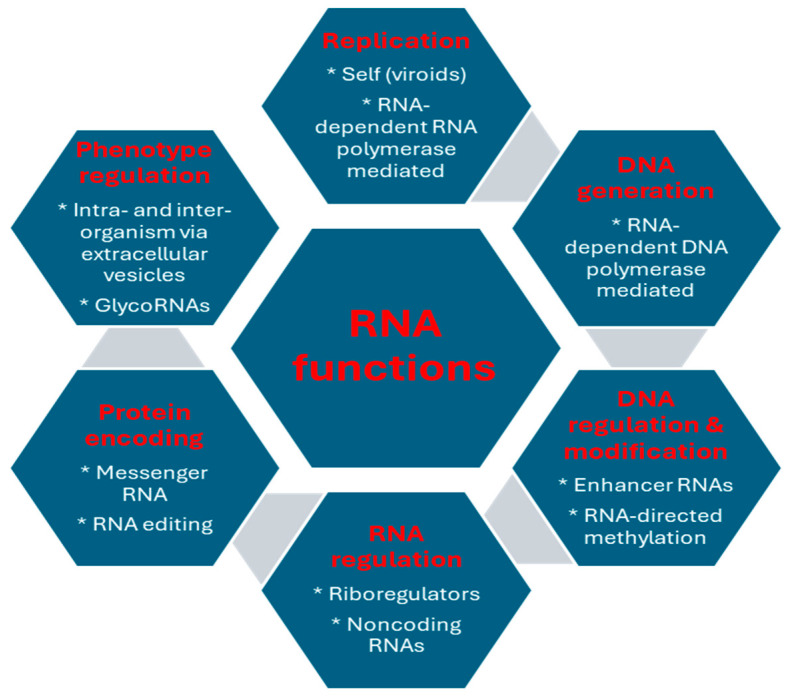
Examples of RNA functions in genetic information flow and biological diversity.

**Table 1 ijms-25-12695-t001:** Some prominent findings in the central molecular biology dogma expansion.

RNA as a genetic information carrier	RNA can store vertically transmitted genetic information and serve as a template to generate DNA. Plant RNA can modify DNA through methylation.
Noncoding RNAs	Over 95% of the genome encodes non-protein-coding RNAs, which play crucial roles in gene expression regulation, chromatin remodeling, and various cellular processes.
RNA diversity and biological complexity	RNA diversity, generated through alternative splicing, polyadenylation, and other means, underlies most intra- and interspecies diversity, influencing protein architecture and disease-linked variation. Antisense strand transcription of some genes and gene expression regulatory elements can generate RNAs and proteins.
RNA modifications	Various RNA modifications, including N6-methyladenosine (m^6^A), directly impact protein production and gene expression regulation, contributing to diverse biological processes and disease mechanisms. RNA editing allows the production of multiple proteins from one gene.
RNA’s extracellular roles	RNA can affect extracellular biology and pathology, such as glycoRNAs influencing immunity and pathogenesis, and RNAs transferred via extracellular vesicles affecting cell biology and interspecies interactions.
Prions and genetic information	Prions, infectious proteins that transmit genetic information without DNA mediation, add complexity to the central dogma by converting normal proteins into misfolded forms, leading to neurodegenerative diseases.
Epigenetic inheritance	Proteins can regulate DNA gene expression via transcription factors and histone modification, underlying epigenetic inheritance beyond genomic DNA inheritance and affecting gene expression and phenotype.
Genetic mosaicism	Many organisms exhibit genetic mosaicism, where transpositions and mutations arising during cell division become fixed in daughter cells, potentially affecting tissue function and disease development.

## Data Availability

Not applicable.

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
