# Peer review of "The RNA Revolution in the Central Molecular Biology Dogma Evolution"

_ijms, 2024, doi:10.3390/ijms252312695_

Round 1
Reviewer 1 Report
Comments and Suggestions for Authors
Haseltine and Patarca succeeded in overlooking the new RNA world and in concisely summarising biological function and regulation of each RNA molecular class. I also appreciate their explanation on fluctuation of the central dogma. It is wonderful and interesting. This review article will not only attract curiosity of scientists but also be highly useful for the educational purpose.
I recommend the manuscript be published.
I have just one request. If the authors provided something like a graphical abstract illustrating the current RNA world, it would make the article absolutely admirable.
Author Response
Thank you to the reviewer for the time and effort to review our manuscript and for the comment to improve its quality. Accordingly, we added Figure 4 with examples of RNA functions in genetic information flow and biological diversity. We also added subsequent paragraphs to emphasize the applications of the RNA revolution. We had submitted a graphical abstract of the expansion of the central molecular biology dogma that was published with the preprint but could not be included in the paper.
The additions are as follows:
Figure 4 summarizes the contributions of RNA to genetic information flow and biological diversity.
Figure 4 can be found in the revised manuscript (this server did not allow to copy-paste it).
The RNA revolution is increasing our understanding of cellular function diversity. For instance, deep RNA sequencing of individual nerve cells has distinguished sixteen neuron types involved in the human sense of touch, similar among mice, macaques, and humans [210]. The interdependence among all biomolecules including nucleic acids, proteins, polysaccharides, lipids, and metabolites warrants further study to better design RNA-based technology for medicine, agriculture, and industry. Improvements in RNA design and delivery technologies for achieving cell type and organ specificity and minimizing off-target effects will increase the effectiveness of RNA-based approaches. The same is true for improving stability, translation, delivery, and immunogenicity avoidance of RNA-based interventions.
Emerging RNA-based approaches are or have the potential to transform human and veterinary medicine, agriculture, and industry, as did the advent of recombinant DNA in the 1980s, creating and disrupting novel diagnostics, preventive approaches, and therapeutics, as evinced by the success of the RNA-based coronavirus vaccines in the recent pandemic.
Applications in the human and veterinary medicine front include RNA vaccines, including RNA-based domestic animal vaccines for microbial and pest resistance; silencing RNAs; RNAs for gene therapy, including gene knockout and knockdown; RNA as CRISPR-Cas guides, and regulators of gene expression; RNAs as antibody and protein therapeutic delivery vectors; RNA aptamers; RNA-based screening, surveillance, diagnostics, prognostication, and personalized medicine; and RNAs for faster growth of food animals, improved animal breeds, and aquaculture.
Agricultural applications include silencing RNAs and messenger RNA modifications to enhance crop yields; RNA interference to introduce favorable traits, such as non-allergic nuts and peanuts, decaffeinated coffee, nutrient-fortified grains, and longer vegetable and fruit shelf life and decreased oxidation when cut; and RNA technologies to increase resistance to pests, drought, salinity, and temperature.
Industrial uses of RNA include RNA-based synthetic biology for the production of biofuels by microorganisms, including RNA interference to favor biofuel production pathways in microbes, and RNA devices to sense biofuel precursors and trigger biofuel production pathways; manufacturing of biodegradable plastics to substitute thermoplastics with environmental benefits, including use of biobased monomers to produce bioplastics with fermentation technology; manufacturing of biodegradable polymers for drug delivery and tissue engineering, and bioscaffolds for in vitro cell culture, organoids, and in vivo implants; and RNA technology for bioremediation of the environment. However, accessing RNA technology in low- and middle-income countries remains challenging.
Reviewer 2 Report
Comments and Suggestions for Authors
The manuscript focuses on the importance of the RNA revolution in overcoming the central dogma of molecular biology. All the elements that contribute to giving RNA a central role are specifically treated and the processes that lead to RNA diversity are described in detail. This part is well-written, and a lot of information contributes to the topic. However, to give greater impetus to the discussion and make it more appealing, even more current implications could be included to understand how considerations of this type can contribute to scientific progress, maybe by inserting specific paragraphs.
Author Response
Thank you for your time and effort in reviewing our manuscript and your comments to improve it. Accordingly, in the last section of the review, we expanded one paragraph and added three more. The end of the manuscript now reads:
The RNA revolution is increasing our understanding of cellular function diversity. For instance, deep RNA sequencing of individual nerve cells has distinguished sixteen neuron types involved in the human sense of touch, similar among mice, macaques, and humans [210]. The interdependence among all biomolecules including nucleic acids, proteins, polysaccharides, lipids, and metabolites warrants further study to better design RNA-based technology for medicine, agriculture, and industry. Improvements in RNA design and delivery technologies for achieving cell type and organ specificity and minimizing off-target effects will increase the effectiveness of RNA-based approaches. The same is true for improving stability, translation, delivery, and immunogenicity avoidance of RNA-based interventions.
Emerging RNA-based approaches are or have the potential to transform human and veterinary medicine, agriculture, and industry, as did the advent of recombinant DNA in the 1980s, creating and disrupting novel diagnostics, preventive approaches, and therapeutics, as evinced by the success of the RNA-based coronavirus vaccines in the recent pandemic.
Applications in the human and veterinary medicine front include RNA vaccines, including RNA-based domestic animal vaccines for microbial and pest resistance; silencing RNAs; RNAs for gene therapy, including gene knockout and knockdown; RNA as CRISPR-Cas guides, and regulators of gene expression; RNAs as antibody and protein therapeutic delivery vectors; RNA aptamers; RNA-based screening, surveillance, diagnostics, prognostication, and personalized medicine; and RNAs for faster growth of food animals, improved animal breeds, and aquaculture.
Agricultural applications include silencing RNAs and messenger RNA modifications to enhance crop yields; RNA interference to introduce favorable traits, such as non-allergic nuts and peanuts, decaffeinated coffee, nutrient-fortified grains, and longer vegetable and fruit shelf life and decreased oxidation when cut; and RNA technologies to increase resistance to pests, drought, salinity, and temperature.
Industrial uses of RNA include RNA-based synthetic biology for the production of biofuels by microorganisms, including RNA interference to favor biofuel production pathways in microbes, and RNA devices to sense biofuel precursors and trigger biofuel production pathways; manufacturing of biodegradable plastics to substitute thermoplastics with environmental benefits, including use of biobased monomers to produce bioplastics with fermentation technology; manufacturing of biodegradable polymers for drug delivery and tissue engineering, and bioscaffolds for in vitro cell culture, organoids, and in vivo implants; and RNA technology for bioremediation of the environment. However, accessing RNA technology in low- and middle-income countries remains challenging.